# Cellular Automaton Simulation of Corrosion in 347H Steel Exposed to Molten Solar Salt at Pilot-Plant Scale

**DOI:** 10.3390/ma18030713

**Published:** 2025-02-06

**Authors:** Juan C. Reinoso-Burrows, Marcelo Cortés-Carmona, Mauro Henríquez, Edward Fuentealba, Andrés Alvear, Carlos Soto, Carlos Durán, Raúl Pastén, Luis Guerreiro, Felipe M. Galleguillos Madrid

**Affiliations:** 1Centro de Desarrollo Energético de Antofagasta, Universidad de Antofagasta, Av. Universidad de Antofagasta 02800, Antofagasta 1271155, Chile; marcelo.cortes@uantof.cl (M.C.-C.); or mauro.henriquez@ciiae.org (M.H.); edward.fuentealba@uantof.cl (E.F.); andres.alvear@uantof.cl (A.A.); carlos.soto@uantof.cl (C.S.); carlos.duran@uantof.cl (C.D.); raul.pasten@uantof.cl (R.P.); felipe.galleguillos.madrid@uantof.cl (F.M.G.M.); 2Iberian Centre for Research in Energy Storage (CIIAE), 10003 Cáceres, Spain; 3ICT—Institute of Earth Sciences, University of Évora, 7000-308 Évora, Portugal; lguerreiro@uevora.pt

**Keywords:** cellular automata, corrosion, solar energy, thermal storage, modeling, CSP

## Abstract

The fast-paced depletion of fossil fuels and environmental concerns have intensified interest in renewable energies, with dispatchable solar energy emerging as a key alternative. Concentrated solar power (CSP) technology, utilizing thermal energy storage (TES) systems with molten salts at 560 °C, offers significant potential for large-scale energy generation. However, these extreme conditions pose challenges related to material corrosion, which is critical for maintaining the efficiency and lifespan of CSP. This research modeled the corrosion process of 347H stainless steel in molten solar salt (60% NaNO_3_ + 40% KNO_3_) melted at 400 °C using a cellular automaton (CA) algorithm. The CA model simulated oxide growth under pilot-plant conditions in a lattice of 400 × 400 cells. SEM-EDS imaging compared the model with a mean squared error of 2%, corresponding to a corrosion layer of 4.25 µm after 168 h. The simulation applied von Neumann and Margolus neighborhoods for the ion movement and reaction rules, achieving a cell size of 0.125 µm and 10.08 s per iteration. This study demonstrates the CA model’s effectiveness in replicating corrosion processes, offering a tool to optimize material performance in CSP systems. Additionally, continuing this investigation could contribute to the development of industrial applications, enabling the design of preventive strategies for large-scale operations.

## 1. Introduction

The energy transition and the high demand for electricity consumption bring a series of challenges and possible problems, especially environmental ones, due to the methods of production used. To meet the growing energy demand and comply with current environmental regulations, new methods of electricity production have been developed, such as solar energy. In recent times, concentrated solar power (CSP) plants have become a promising technology for large-scale electricity production. This technology uses tanks that store thermal energy by employing heat transfer fluid (HTF); this energy is used during the night to produce electricity, thereby maintaining a continuous production cycle [1]. The most commonly used heat transfer medium is a binary mixture of nitrate salts, known as solar salt, composed of 60% sodium nitrate and 40% potassium nitrate [1]. Storing thermal energy using molten salts creates extreme operating conditions, reaching over 560 °C, a temperature that affects the materials used in the construction of storage tanks [2,3]. These conditions can cause corrosion damage to the steel used, potentially leading to catastrophic problems for the environment, the plant, or the working personnel [4,5,6,7].

Techniques for studying corrosion are valuable and provide critical information [8,9,10]. However, they often require several weeks to complete. To address this, the use of the cellular automata (CA) model was proposed as an alternative for evaluating the corrosion behavior of steel used in thermal storage systems with molten salts [11,12,13]. This approach aims to determine the corrosion rate (C.R.) and simulate oxide formation under operating conditions, using a pilot plant at the University of Antofagasta as the study object.

The CA model has been employed by various authors to describe the behavior of natural phenomena, including steel corrosion at different temperatures [14,15,16,17]. Notably, Reinoso-Burrows et al. [18] conducted a comprehensive literature review on the application of the CA model, emphasizing its ability to simulate diverse patterns in the field of corrosion. However, no studies have been reported that utilize this model under pilot-plant operating conditions, particularly with the same combination of 347H stainless steel, molten solar salt, and a temperature of 400 °C, as this work uniquely sought to achieve. This study represented a novel approach by extending the CA model’s application to real-world operational scenarios, bridging the gap between theoretical modeling and practical industrial use.

For this reason, the objectives of this study were to use the CA model to simulate the formation of corrosion products on 347H stainless steel exposed to molten solar salt at 400 °C and to compare the results from an experimental procedure. To achieve this, Fe-Cr-Ni elements, representing the main components of the studied steel, were considered. Various corrosion mechanisms for molten nitrate salts have been proposed for these elements. However, most authors have attributed corrosion primarily to nitrate reduction (reaction 1) [7,19,20,21], while dissolved oxygen in the salt has been reported to accelerate corrosion through oxygen reduction (reaction 2) [22].NO_3_^−^ + 2e^−^ ↔ NO_2_^−^ + O^2−^(1)O_2_ + 4e^−^ ↔ 2O^2−^(2)

This leads to a series of reactions that govern the corrosion process (reactions 3 to 5). Similarly, the oxide ions present in the molten salt interact not only with iron but also with other key elements in the composition of stainless steel, such as chromium and nickel, as proposed by Mallco et al. [19].Fe + O^2−^ → FeO + 2e^−^(3)3FeO + O^2−^ → Fe_3_O_4_ + 2e^−^(4)2Fe_3_O_4_ + O^2−^ → Fe_2_O_3_ + 2e^−^(5)2Cr + O^2−^ → 3Cr_2_O_3_ + 6e^−^(6)2Ni + O^2−^ → 2NiO + 2e^−^(7)

The present work considered the corrosion mechanism with the above reactions as input data for the model.

## 2. Methodology: Experimental Procedure and Modeling

### 2.1. Immersion Test

The experimental procedure for validating the simulated results involved an immersion test, which included the preparation of 347H stainless steel samples (Table 1). Rectangular samples were cut to dimensions of 300 mm in length, 100 mm in width, and 100 mm in thickness. An 8 mm diameter hole was drilled into each sample to be placed on the designed rods (Figure 1) for introduction into the pilot molten salt tank (Figure 2).

The samples underwent sanding and polishing treatments up to a 3000-grit size and were weighed in triplicate on an ME204T/00 analytical balance (Mettler-Toledo, Columbus, OH, USA) with a precision of ± 0.1 mg before exposure.

After exposure, each sample was removed from the hot tank at 168 h and allowed to cool slowly to room temperature. Excess salt was then removed by rinsing with hot distilled water, acetone, and ethanol [23].

The salt mixture used was a solar salt (60% NaNO_3_ and 40% KNO_3_) from SQM whose composition is shown in Table 2.

The samples were immersed in 700 kg of solar salt within the thermal storage tank located at the University of Antofagasta, Chile. The salt was melted using four 1200-watt electric heaters, reaching an operating temperature of 400 °C. The storage system included a pump that circulated the molten salts within the tank, enabling studies under real operating conditions.

For the SEM/EDS analysis, the cross-sectional corrosion layer of the sample exposed to solar salt needed to be observed. For this purpose, post-exposure the sample was immersed in a granular resin heated to 100 °C, which fused with the sample. Subsequently, one side was polished to reveal the steel and the corrosion layer.

### 2.2. Modeling

As part of the methodology, the modeling consisted of the following phases: the corrosion mechanism, site definition, the model parameters, and the transformation rules.

### 2.3. Corrosion Mechanism

The proposed mechanism considers reactions 3, 4, 5, 6, and 7, involving an oxyanion salt. Due to the temperature, the nitrate dissociates, generating the oxide ion (reaction 1). Based on these reactions, various sources of oxide ions are generated. Additionally, this study proposes an extra source corresponding to the reduction of molecular oxygen (reaction 2), contributing to the concentration of oxide ions in the molten salt mass.

In the long term, only hematite, magnetite, chromium oxide, and nickel oxide elements are considered in the model from the proposed mechanism because the model does not consider intermediate reactions, and in the long term the presence of FeO at high temperatures is not reported. Because of this, the reactions considered are the following:3Fe + 4O^2−^ → Fe_3_O_4_(8)2Fe_3_O_4_ + O^2−^ → 3Fe_2_O_3_(9)2Cr + 3O^2−^ → Cr_2_O_3_(10)Ni + O^2−^ → NiO(11)

### 2.4. Site Definition

A two-dimensional grid of cells must be defined to simulate a corrosion process using cellular automata. The grid size will depend on the data processing capacity and the number of rules to be used. Also, the grid size must allow for adequate identification of the different elements of the process. Considering these aspects, a grid of 400 × 400 cells was chosen in this study. On the other hand, to facilitate the process of understanding the methodology, letters were assigned to each cell of the grid, representing the compounds and elements involved in the corrosion process, as shown in Figure 3, with the corresponding letters displayed in Table 3. All elements were considered fixed, except for site B, which can move throughout the grid. The initial concentrations of sites A, E, and G (representing the elements Fe, Cr, and Ni, respectively) were 73.5%, 17.5%, and 9%, respectively. The oxide ion was represented by site B, while corrosion sites C, D, F, and H, correspond to the compounds Fe_3_O_4_, Fe_2_O_3_, Cr_2_O_3_, and NiO, respectively, were generated throughout the simulation.

The proposed corrosion mechanism, combined with labeling the elements and compounds involved in the simulation, served as the foundation for the model’s simplified equations. This approach was designed to optimize the computation time and streamline the analysis of the model’s results.A + B → C(12)C + B → D(13)E + B → F(14)G + B → H(15)

### 2.5. Model Parameters

Site B was considered the main corrosive agent in the model. Its movement utilized the von Neumann neighborhood and could move randomly across the entire grid, while the reaction rules used the Margolus neighborhood to reduce programming times as shown in Figure 4a and Figure 4b respectively.

Movement probabilities were assigned to site B only when it was adjacent to any site F, corresponding to chromium oxide, representing the chromium passivation elements seen in Figure 5. The probabilities were Pd4 < Pd3 < Pd2 < Pd1 < Pd0, corresponding to 0.2, 0.4, 0.6, 0.8, and 1.0, respectively. Additionally, a constant concentration of 25% was considered in each iteration. In this study, this constant concentration at site B implied continuous renewal of the oxide ion. It was assumed that there was no oxygen depletion because the tank had a lid for specimen access, allowing atmospheric air renewal.

### 2.6. Transformation Rules

Once site B completed its movement, the Margolus neighborhood was applied to determine the reaction and transformation rules. Seventeen possible cases were defined according to each site on the grid. Figure 6 shows the five possible cases for site A, with each possibility having a probability of occurrence based on the relative probability of oxidation, which considers the standard reduction potential using equation 16, whose results are shown in Table 4.(16)$$P.ox\_rel=e^{-E{}^{\circ}}$$

## 3. Results

### 3.1. SEM/EDS

The SEM cross-section results are shown in Figure 7. The sample was exposed to a temperature of 400 °C for a continuous period of 168 h in solar salt.

The EDS mapping is shown in Figure 8, which displays the concentration of Cr (letter b), an important element in the corrosion resistance of 347H stainless steel, and two other simulated elements, iron and nickel. Black areas indicate zones with low or no chromium concentration, while fluorescent green highlights areas with the presence of chromium. From left to right in the image, a transition can be observed from low or no chromium concentration to an increase in the chromium concentration within the steel, as shown on the right side of the image.

### 3.2. Modeling

Initially, 100,000 iterations were simulated. Figure 9 shows the general behavior of evolution in the concentration of all the sites in the model. As the simulation progressed, the concentration of metallic sites A (Fe), E (Cr), and G (Ni) decreased, and due to the reaction and transformation rules, they were converted into corrosion products C, D, F, and H, which increased their concentration over time, as shown in Figure 10.

Figure 11 shows the concentrations of simulated sites at 60,000 iterations with respect to the metal depth, which corresponds to 200 cells. The *X*-axis represents the steel depth in terms of the cell count, while the *Y*-axis indicates the sums of the sites for the different simulated elements. Initially, there was a lower concentration of sites A, E, and G up to approximately 34 cells, which marked the end of the corrosion layer. Furthermore, in areas with reduced concentrations of Fe, Cr, and Ni due to the corrosion process, the model generated sites C, D, F, and H, which corresponded to the simulated corrosion products.

Figure 12 shows a cross-section of the model and the evolution of the corrosion layer throughout the simulation.

### 3.3. Data Processing

Data processing was carried out to simulate results with respect to the experimental EDS results shown in Figure 13. The mean squared error between the simulated chromium curve and the EDS data was calculated, resulting in a 2% error at 60,000 steps, the lowest among all simulation times. This corresponded to 168 h of exposure, with each iteration representing 10.08 s and a cell size of 0.125 µm.

However, the results shown in Figure 14 and Figure 15, corresponding to the elements nickel and iron, respectively, which were obtained using the same methodology, indicate a slight difference in the curve fitting provided by the model. This difference is primarily due to the element concentration rather than the corrosion layer zone relative to the steel depth.

## 4. Conclusions

The results obtained through SEM/EDS allowed the identification of areas with lower concentrations of key elements involved in the corrosion process, which was fundamental for providing experimental validation of the simulated results. The key element considered in this model is chromium, as the formation of chromium oxide plays a central role in controlling the corrosion process and rate, surpassing the influence of hematite, magnetite, or nickel oxide. Once the curve was adjusted to chromium, the results indicated that each iteration corresponded to 10.08 experimental seconds, and each cell in the mesh represented 0.125 µm, with a corrosion layer thickness of 4.25 µm in the studied steel. The model achieved an error of <2% at 60,000 iterations. Although the model was refined to achieve a low error rate, further validation and adjustments could potentially reduce it even further.

The slight mismatch between the iron and nickel curves in the uncorroded areas of the model compared to the EDS results may suggest the need to adjust the concentration in the simulated mesh matrix. Additionally, this discrepancy between the simulation and the EDS graphs is smaller in the corrosion product region, indicating a good correlation with the model in that area.

This investigation represents progress in simulating the corrosion process in thermal storage systems with molten salts. The results indicate that the cellular automaton model can replicate the formation of expected corrosion products under the operating conditions of a pilot plant. However, a significant gap remains to be addressed regarding the use of the CA model. One challenge is to simulate more complex corrosion mechanisms, and another is to obtain a larger amount of experimental data at the pilot-plant scale. Additionally, for this model, it is important to use high-resolution images, which represent an opportunity for improvement in the continuation of this work. Concerning the transfer of knowledge, data investigated in this study were used as content in a virtual reality concept, which should facilitate the transmission of knowledge as well as enhance the dissemination of the results.

## Figures and Tables

**Figure 1 materials-18-00713-f001:**
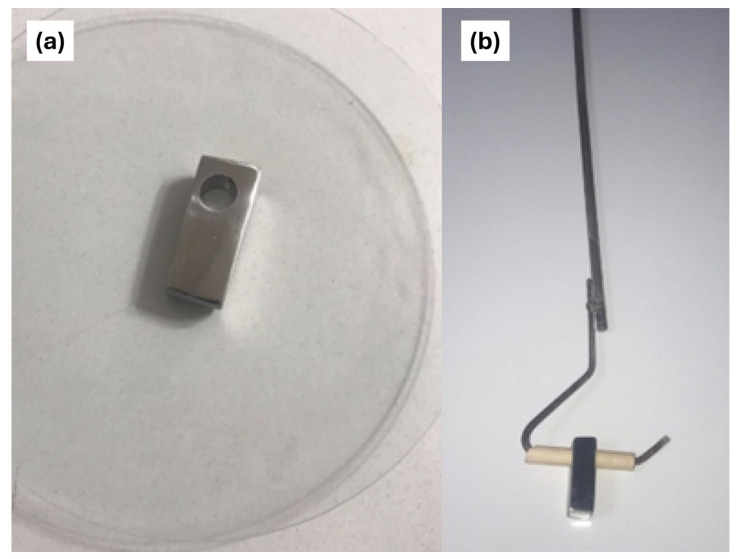
(**a**) Coupon and (**b**) coupon holder designed for the pilot plant.

**Figure 2 materials-18-00713-f002:**
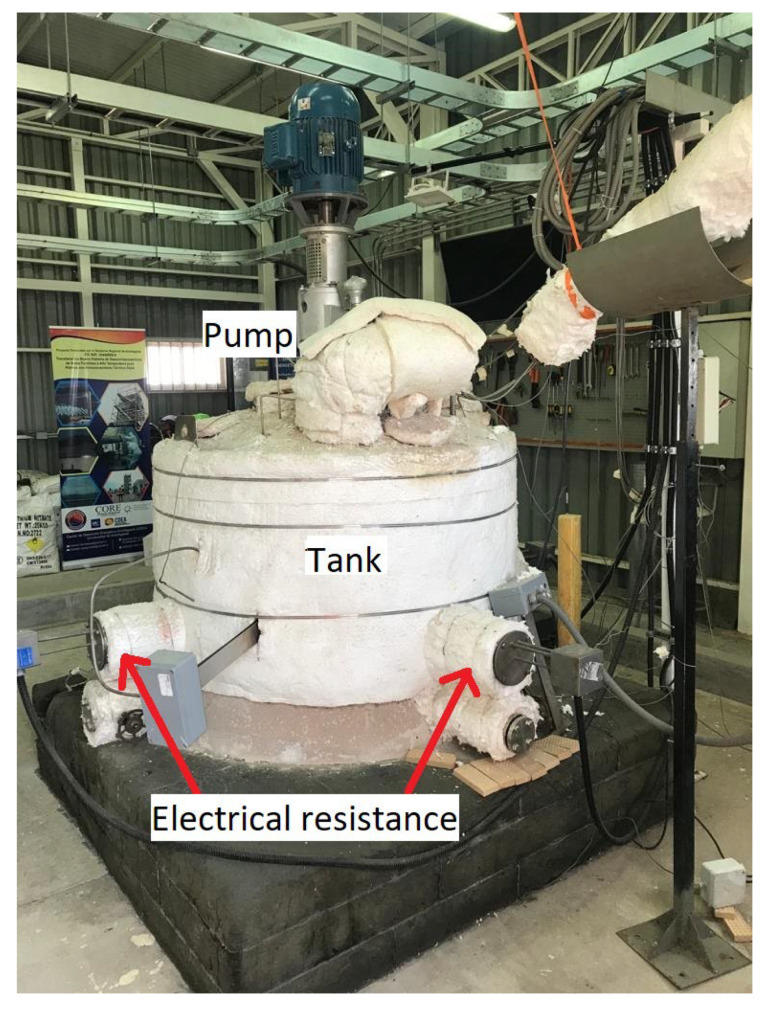
Molten-salt thermal storage pilot plant.

**Figure 3 materials-18-00713-f003:**
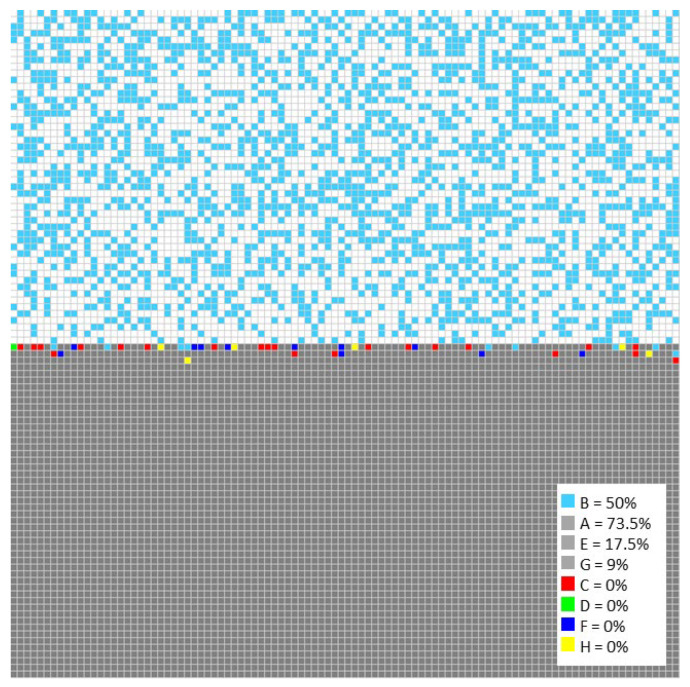
Distribution of sites in cellular automata matrix.

**Figure 4 materials-18-00713-f004:**
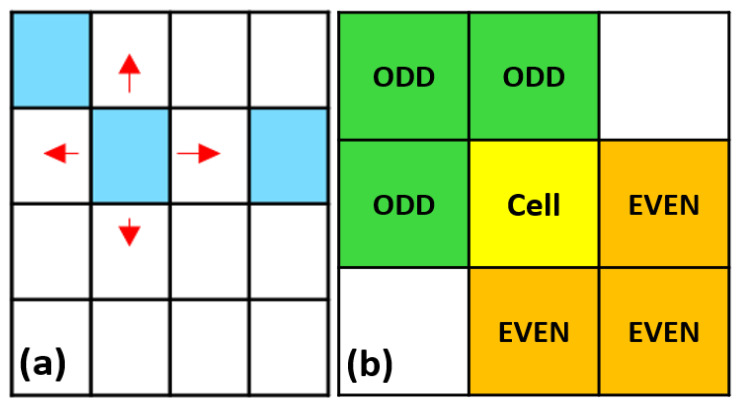
(**a**) Von Neumann and (**b**) Margolus neighborhoods. In (**a**), the red arrows indicate the possible movement directions of site B in each iteration, occurring in only one direction at a time. The colors are merely representative and indicate cells occupied by a site.

**Figure 5 materials-18-00713-f005:**
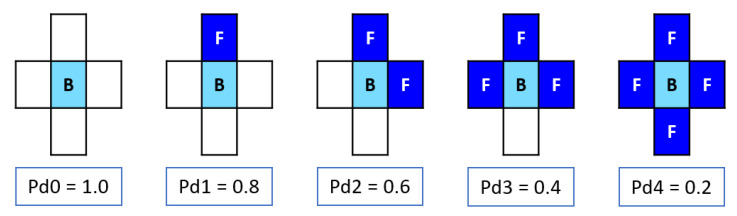
Probability of motion for site B. The light blue color indicates site B, while the dark blue color represents site F.

**Figure 6 materials-18-00713-f006:**
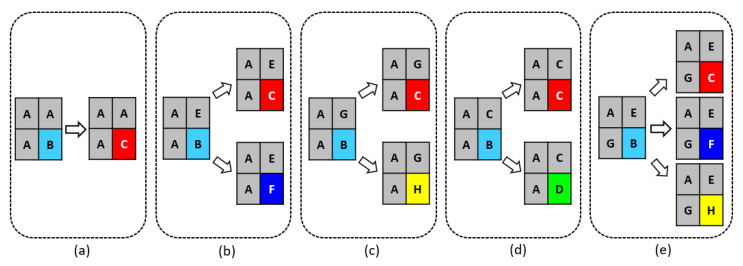
Reaction and transformation rules for site A. For case (**a**), site B has only one option to transform into site C. For cases (**b**–**e**), each site B can transform into only one site according to the indicated arrow, based on Equations (12)–(16).

**Figure 7 materials-18-00713-f007:**
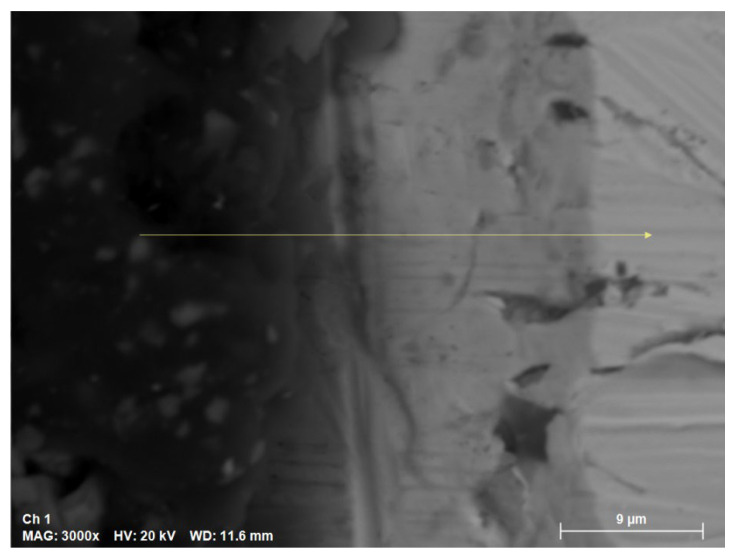
SEM cross-section of sample after 168 h. The yellow line indicates the detection zone of the elements used for model validation.

**Figure 8 materials-18-00713-f008:**
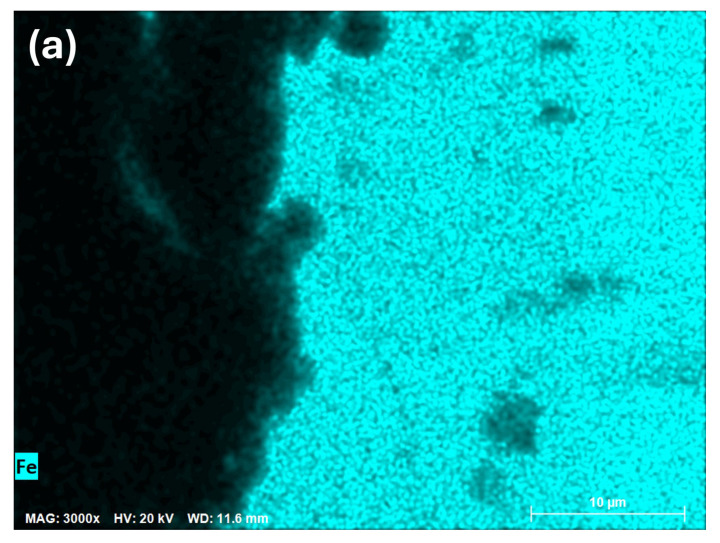
Mapping EDS cross-sections of (**a**) iron, (**b**) chromium, and (**c**) nickel in sample after 168 h.

**Figure 9 materials-18-00713-f009:**
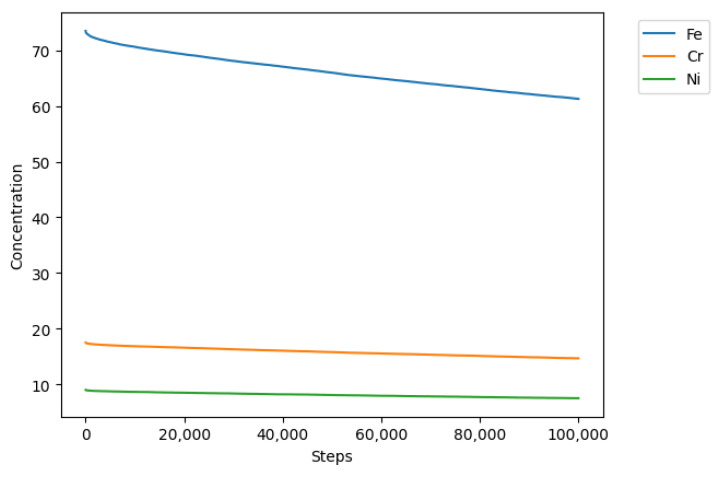
Concentrations of sites A, E, and G after 100,000 iterations with 25% B sites.

**Figure 10 materials-18-00713-f010:**
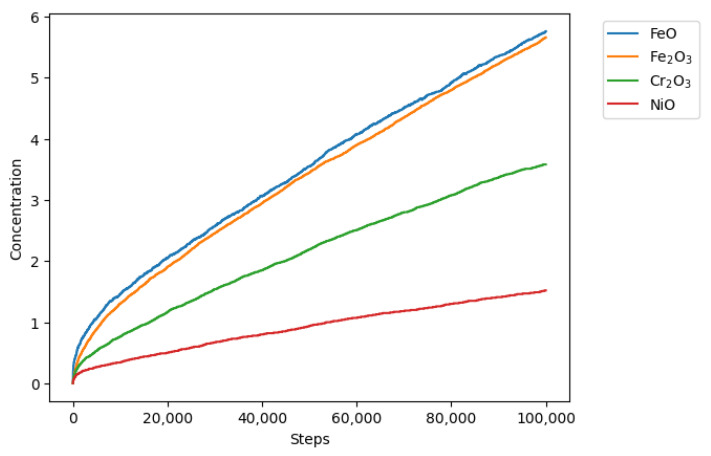
Concentrations of sites C, D, F, and H after 100,000 iterations with 25% B sites.

**Figure 11 materials-18-00713-f011:**
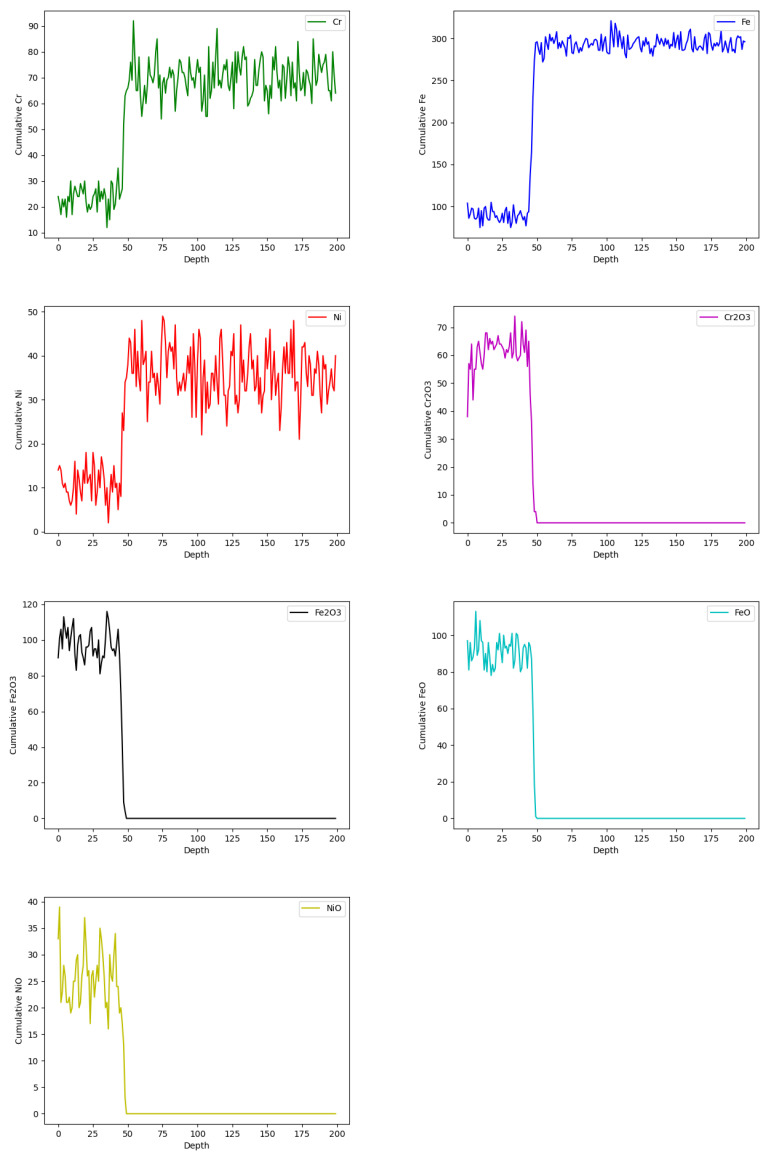
Evolution of the concentrations of the simulated elements at 60,000 iterations.

**Figure 12 materials-18-00713-f012:**
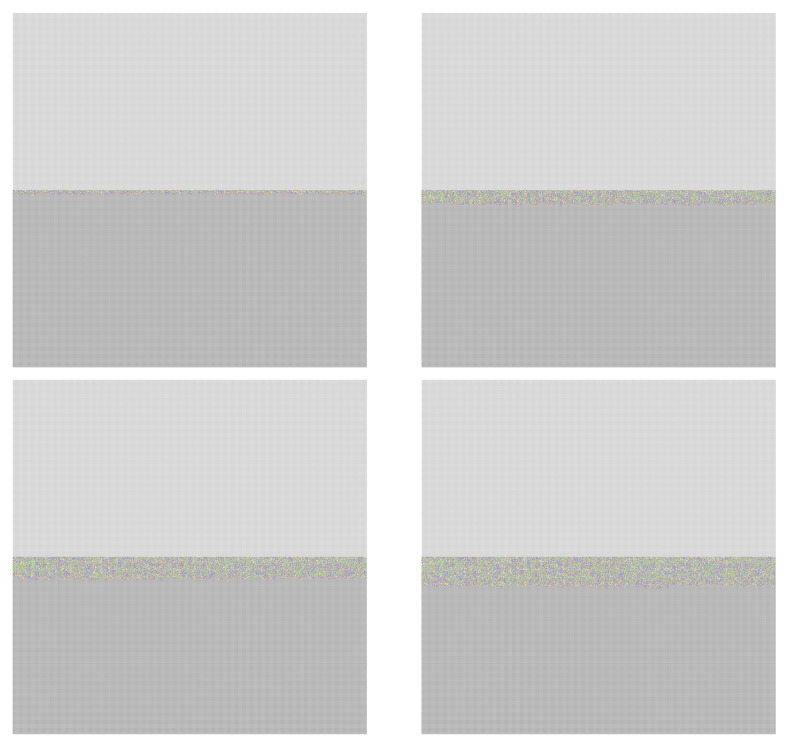
A cross-sectional view of the corrosion layer growth at 1,000, 20,000, 40,000, and 60,000 steps.

**Figure 13 materials-18-00713-f013:**
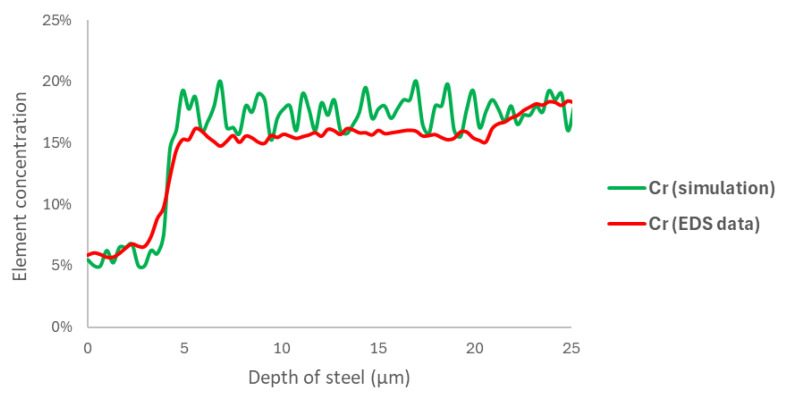
A comparison of the EDS curve for the 168 h sample and the chromium site concentration at 60,000 steps.

**Figure 14 materials-18-00713-f014:**
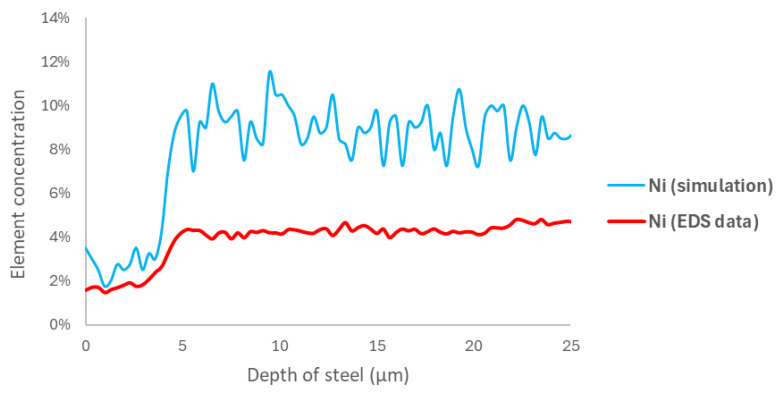
A comparison of the EDS curve for the 168 h sample and the nickel site concentration at 60,000 steps.

**Figure 15 materials-18-00713-f015:**
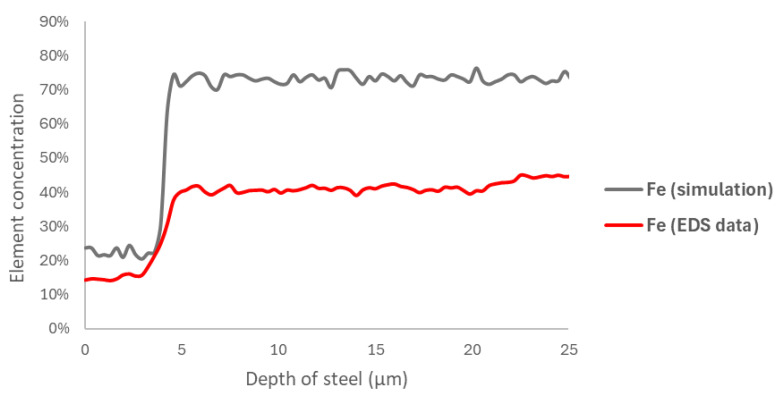
A comparison of the EDS curve for the 168 h sample and the iron site concentration at 60,000 steps.

**Table 1 materials-18-00713-t001:** Chemical composition of 347H SS.

Chemical Composition of AISI 347H Stainless Steel [%p/p]											
Fe	C	Si	Mn	Cr	Mo	Ni	Al	Co	Cu	Nb	W
70.5	0.04565	0.1885	1.735	17.05	0.396	8.96	0.0068	0.114	0.4195	0.4015	0.0378

**Table 2 materials-18-00713-t002:** Chemical composition of solar salt.

Chemical Composition of Solar Salt	
Compound	[%p/p]
NaNO_3_	60.2
KNO_3_	39.8

**Table 3 materials-18-00713-t003:** Labeling of the elements and compounds involved in the simulation.

Fe	O^2−^	Fe_3_O_4_	Fe_2_O_3_	Cr	Cr_2_O_3_	Ni	NiO
A	B	C	D	E	F	G	H

**Table 4 materials-18-00713-t004:** Relative probability according to standard reduction potential.

Site	Element	E° (V)	P.ox_rel	P.rel₁ (%)	P.rel₂ (%)	P.rel₃ (%)	P.rel₄ (%)
A	Fe	−0.44	1.55	31.4	57.4	-	54.6
E	Cr	−0.74	2.09	42.4	42.6	61.8	-
G	Ni	−0.25	1.29	26.2	-	38.2	45.4

## Data Availability

The original contributions presented in this study are included in this article. Further inquiries can be directed to the corresponding author.

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
