# Peer review of "Cellular Automaton Simulation of Corrosion in 347H Steel Exposed to Molten Solar Salt at Pilot-Plant Scale"

_materials, 2025, doi:10.3390/ma18030713_

Round 1

Reviewer 1 Report

Comments and Suggestions for Authors

The aim of this work is to model the corrosion process of 347H stainless steel in molten solar salt at 400°C using a cellular automaton (CA) algorithm. The model simulates oxide growth and SEM-EDS imaging experimentally verify the modelling results.

The validation of the simulated putputs has been made by measuring the oxide layer by SEM analysis.

What about running a descaling ISO procedure to weight the oxide layer formed on the specimens?

The experimental validation would be more effective by eventually follow the kinetic of formation during time, meaning sampling specimens immersed in molten salt in diffeter time intervals (200-500-1000 h). It is in your plans?

Author Response

Comments 1: The aim of this work is to model the corrosion process of 347H stainless steel in molten solar salt at 400°C using a cellular automaton (CA) algorithm. The model simulates oxide growth and SEM-EDS imaging experimentally verify the modelling results.
The validation of the simulated outputs has been made by measuring the oxide layer by SEM analysis.
What about running a descaling ISO procedure to weight the oxide layer formed on the specimens?.

Response 1: Thank you for the suggestion. In this study, the oxide layer was primarily analyzed using SEM-EDS to identify the composition and distribution of corrosion products. This methodology was chosen due to technical constraints in our pilot plant system, where the number of specimens that can be submerged simultaneously is limited. These limitations prompted us to focus on SEM-EDS analysis for this phase of the study.
We are currently working on modifying the specimen system in the pilot plant to allow for a larger number of simultaneous samples. This improvement will enable us to incorporate additional procedures, such as the descaling of the oxide layer in future experiments.

Comments 2: The experimental validation would be more effective by eventually follow the kinetic of formation during time, meaning sampling specimens immersed in molten salt in diffeter time intervals (200-500-1000 h). It is in your plans?.

Response 2: Thank you for your observation. We agree and what you mention would significantly enhance the experimental validation of the study. This approach is indeed within our plans for future work.

Currently, our pilot plant system is limited in its capacity to simultaneously submerge multiple specimens, which constrained our ability to perform time-interval studies with sufficient replicates. Although we conducted preliminary tests at different time intervals, these results were not included in the article as they could not be performed in triplicate, and we prioritized ensuring the reliability of the data presented.

Reviewer 2 Report

Comments and Suggestions for Authors

This paper mainly introduces a model and method for the molten salt corrosion of 347H stainless steel, and demonstrates the feasibility of this approach through experimental data. Here are my comments:

1.    In Methodology: Experimental and modeling (Section 2), the sample size is 300mm × 100mm × 100mm. It would be quite challenging to grind, polish, and conduct SEM/EDS observations on such a rod-shaped object. Did the authors cut a small piece from the corroded rod samples for experimental analysis? Could the cutting process introduce new uncertainties? Authors should provide more details on the experimental procedures.

2.    Please specify the oxygen environment in which the experimental setup was conducted, including the oxygen solubility.

3.    In Figure 4, the schematic diagrams are not labeled with (a) and (b); also, the thousands digit identifiers for 100,000 and 60.000 are not consistent. Please check the manuscript to avoid such minor errors.

4.    The manuscript mentions several important elements involved in the reaction, which are Fe, Cr, Ni, and O. The manuscript only confirms that the distribution of Cr in the model is consistent with the SEM/EDS data. However, the manuscript does not provide simulation results for Fe, Ni, and O elements. Providing data for Fe, Ni, and O, for example, data similar to the EDS data in Figure 8 and the comparative data in Figure 13, and it could greatly enhance the feasibility of the models and methods used in the paper.

Author Response

Comments 1: In Methodology: Experimental and modeling (Section 2), the sample size is 300mm × 100mm × 100mm. It would be quite challenging to grind, polish, and conduct SEM/EDS observations on such a rod-shaped object. Did the authors cut a small piece from the corroded rod samples for experimental analysis? Could the cutting process introduce new uncertainties? Authors should provide more details on the experimental procedures.

Response 1: Thank you for your questions. The specimens were cut and polished before being mounted on the rod for exposure. No further cutting was performed afterward. The post-treatment involved embedding the specimens in granular resin to prevent the detachment of corrosion products, followed by polishing one side without additional cutting. As you mentioned, this method might introduce some uncertainty; however, it is the approach used to determine the thickness of the corrosion layer from the perspective of the cross-section. The details are further clarified in the methodology section.

Comments 2: Please specify the oxygen environment in which the experimental setup was conducted, including the oxygen solubility.

Response 2: Thank you for your question. Although the oxygen solubility in molten solar salt was not directly measured in this study, an estimation of the oxide ion concentration was made based on corrosion mechanisms reported in the literature, specifically those involving nitrate dissociation. It is worth noting that the pilot tank has a lid that could allow a slight ingress of air; however, for this analysis, only the oxygen contribution from nitrate dissociation was considered.

Comments 3: In Figure 4, the schematic diagrams are not labeled with (a) and (b); also, the thousands digit identifiers for 100,000 and 60.000 are not consistent. Please check the manuscript to avoid such minor errors.

Response 3: Thank you for your kind comment. Changes were made to Fig. 4, and the identification of digits was corrected.

Comments 4: The manuscript mentions several important elements involved in the reaction, which are Fe, Cr, Ni, and O. The manuscript only confirms that the distribution of Cr in the model is consistent with the SEM/EDS data. However, the manuscript does not provide simulation results for Fe, Ni, and O elements. Providing data for Fe, Ni, and O, for example, data similar to the EDS data in Figure 8 and the comparative data in Figure 13, and it could greatly enhance the feasibility of the models and methods used in the paper.

Response 4: Thank you very much for your kind comment. Indeed, the SEM/EDS analysis of chromium was used as a reference due to its crucial role in the corrosion process, serving as a basis for data processing. However, considering your valuable suggestion, the EDS mapping of iron and nickel has also been included at line 224 to 226 and line 233 to 236.

Reviewer 3 Report

Comments and Suggestions for Authors

The abstract meets the basic requirements by establishing the objective of the study (modeling the corrosion of 347H stainless steel in solar salts using a cellular automaton model) and its context within the interest in renewable energies and CSP technology. The methodology is adequately described, highlighting the use of simulations in a 400x400 cell lattice, validation through SEM-EDS, and the application of specific reaction and ionic movement rules. It also summarizes key results, such as a mean squared error of 2%, a corrosion layer thickness of 4.25 µm, and precise iteration times. Finally, it concludes with the model’s effectiveness in replicating corrosion processes and its potential utility in optimizing materials for CSP systems. However, the practical scope of the study and the direct implications of the results for real-world applications could be further detailed to strengthen the connection between the findings and their impact on CSP system design and maintenance.

The introduction provides the necessary context to situate the reader in the topic and highlight the importance of the article. It adequately presents the background on energy transition, concentrated solar power (CSP) technology, and the challenges related to corrosion in thermal storage systems. Furthermore, it justifies the relevance of the work by highlighting the limitations of traditional corrosion study techniques and proposing the cellular automaton (CA) model as a more efficient alternative. The cited literature appears to be appropriately selected, referencing relevant studies on corrosion mechanisms and previous applications of the CA model.

However, the section does not fully clarify why this specific study is necessary or which particular gap in the literature it addresses. While the CA model's ability to evaluate corrosion patterns and the key chemical reactions are mentioned, it would be beneficial to more explicitly link these points to an unmet scientific or practical need. Additionally, although the article's objective is introduced, it could be stated more directly and connected to the study’s justification to better emphasize its novelty and contribution to the field.

Overall, the introduction largely fulfills its purpose, but it would benefit from a clearer focus on justifying the article’s necessity and highlighting it as a unique contribution within the context of the existing literature.

The methodology section provides sufficient detail for replication, covering both the experimental procedures and the computational modeling aspects. The experimental subsection clearly describes the preparation of stainless steel 347H samples, including their dimensions, chemical composition, and pre-treatment steps such as sanding and polishing. Furthermore, it specifies the composition of the molten solar salt, the test conditions in a thermal storage pilot plant, and the post-exposure analysis protocols, aligning with ISO standards. The use of 700 kg of molten salt at 400°C and the inclusion of a circulation system ensure realistic operating conditions, enhancing the reliability of the results.

The modeling subsection systematically outlines the steps involved in simulating the corrosion process. It begins with a well-defined corrosion mechanism, describing the chemical reactions considered and the long-term assumptions regarding stable corrosion products. The 2D cellular automaton model is presented with detailed information about grid size, site definitions, and initial conditions, facilitating replication by competent researchers. Movement and transformation rules are also explicitly described, including probabilities, neighborhoods used (von Neumann and Margolus), and reaction equations.

While comprehensive, the methodology could benefit from minor clarifications. For example, additional context on how the selected grid size and parameters (e.g., 25% constant concentration of site B) were determined would enhance transparency. Similarly, a brief discussion of the assumptions and limitations of both the experimental and modeling approaches would provide a more balanced perspective.

Results

The SEM cross-sectional images show the material structure after 168 hours of exposure to 400°C in solar salt. However, one of the critical observations is the low resolution of the images, which makes it difficult to accurately visualize the observed features. This could be improved to ensure that important details of the corrosion areas are clearly identifiable. Figure 8 presents the EDS mapping, which shows the concentration of chromium, a key element in the corrosion resistance of stainless steel 347H. Dark areas indicate low or no chromium concentration, while fluorescent green highlights areas with the presence of chromium. The transition from low concentration zones to those with higher chromium concentration is evident in the image.

Regarding the modeling, 100,000 iterations were performed, and the evolution of the site concentration in the model is shown in Figures 9 and 10. As the simulation progresses, the concentration of metallic sites (A, E, G) decreases, and these are converted into corrosion products (C, D, F, H), which increase in concentration. Figure 11 shows the concentration of simulated sites with respect to the metal depth after 60,000 iterations. In this graph, the decrease in Fe, Cr, and Ni concentrations marks the beginning of the corrosion layer, which is clearly visible in the simulation.

Finally, Figure 12 illustrates the evolution of the corrosion layer over the simulation time, showing a cross-sectional view of the layer growth at various steps of the model. This provides a clear image of how corrosion develops as the simulation progresses.

Data processing was carried out to compare the simulated results with the experimental EDS data. The comparison shows that the mean squared error between the simulated curve and the EDS data is 2% at 60,000 steps, which is a relatively low error. This result corresponds to 168 hours of exposure, with each iteration representing 10.08 seconds and a cell size of 0.125 µm.

Author Response

Comments 1: The abstract meets the basic requirements by establishing the objective of the study (modeling the corrosion of 347H stainless steel in solar salts using a cellular automaton model) and its context within the interest in renewable energies and CSP technology. The methodology is adequately described, highlighting the use of simulations in a 400x400 cell lattice, validation through SEM-EDS, and the application of specific reaction and ionic movement rules. It also summarizes key results, such as a mean squared error of 2%, a corrosion layer thickness of 4.25 µm, and precise iteration times. Finally, it concludes with the model’s effectiveness in replicating corrosion processes and its potential utility in optimizing materials for CSP systems. However, the practical scope of the study and the direct implications of the results for real-world applications could be further detailed to strengthen the connection between the findings and their impact on CSP system design and maintenance. 

Response 1: Thank you for pointing this out. We agree with your comment; information has been added from lines 25 to 29.

Comments 2: The introduction provides the necessary context to situate the reader in the topic and highlights the importance of the article. It adequately presents the background on energy transition, concentrated solar power (CSP) technology, and the challenges related to corrosion in thermal storage systems. Furthermore, it justifies the relevance of the work by highlighting the limitations of traditional corrosion study techniques and proposing the cellular automaton (CA) model as a more efficient alternative. The cited literature appears to be appropriately selected, referencing relevant studies on corrosion mechanisms and previous applications of the CA model.

However, the section does not fully clarify why this specific study is necessary or which particular gap in the literature it addresses. While the CA model's ability to evaluate corrosion patterns and the key chemical reactions are mentioned, it would be beneficial to more explicitly link these points to an unmet scientific or practical need. Additionally, although the article's objective is introduced, it could be stated more directly and connected to the study’s justification to better emphasize its novelty and contribution to the field.

Overall, the introduction largely fulfills its purpose, but it would benefit from a clearer focus on justifying the article’s necessity and highlighting it as a unique contribution within the context of the existing literature.

Response 2: Thank you for your kind comment, we hope to provide more complete information to your concerns on lines 57 to 67.

Comments 3: The methodology section provides sufficient detail for replication, covering both the experimental procedures and the computational modeling aspects. The experimental subsection clearly describes the preparation of stainless steel 347H samples, including their dimensions, chemical composition, and pre-treatment steps such as sanding and polishing. Furthermore, it specifies the composition of the molten solar salt, the test conditions in a thermal storage pilot plant, and the post-exposure analysis protocols, aligning with ISO standards. The use of 700 kg of molten salt at 400°C and the inclusion of a circulation system ensure realistic operating conditions, enhancing the reliability of the results.

The modeling subsection systematically outlines the steps involved in simulating the corrosion process. It begins with a well-defined corrosion mechanism, describing the chemical reactions considered and the long-term assumptions regarding stable corrosion products. The 2D cellular automaton model is presented with detailed information about grid size, site definitions, and initial conditions, facilitating replication by competent researchers. Movement and transformation rules are also explicitly described, including probabilities, neighborhoods used (von Neumann and Margolus), and reaction equations.

While comprehensive, the methodology could benefit from minor clarifications. For example, additional context on how the selected grid size and parameters (e.g., 25% constant concentration of site B) were determined would enhance transparency. Similarly, a brief discussion of the assumptions and limitations of both the experimental and modeling approaches would provide a more balanced perspective.

Response 3: Thank you for your kind comment and we accept your correction. Changes were made to the document on lines 155 to 157. Also, on line 187 (Figure 4) and lines 192 to 195.

Comments 4: The SEM cross-sectional images show the material structure after 168 hours of exposure to 400°C in solar salt. However, one of the critical observations is the low resolution of the images, which makes it difficult to accurately visualize the observed features. This could be improved to ensure that important details of the corrosion areas are clearly identifiable. Figure 8 presents the EDS mapping, which shows the concentration of chromium, a key element in the corrosion resistance of stainless steel 347H. Dark areas indicate low or no chromium concentration, while fluorescent green highlights areas with the presence of chromium. The transition from low concentration zones to those with higher chromium concentration is evident in the image.

Regarding the modeling, 100,000 iterations were performed, and the evolution of the site concentration in the model is shown in Figures 9 and 10. As the simulation progresses, the concentration of metallic sites (A, E, G) decreases, and these are converted into corrosion products (C, D, F, H), which increase in concentration. Figure 11 shows the concentration of simulated sites with respect to the metal depth after 60,000 iterations. In this graph, the decrease in Fe, Cr, and Ni concentrations marks the beginning of the corrosion layer, which is clearly visible in the simulation.

Finally, Figure 12 illustrates the evolution of the corrosion layer over the simulation time, showing a cross-sectional view of the layer growth at various steps of the model. This provides a clear image of how corrosion develops as the simulation progresses.

Data processing was carried out to compare the simulated results with the experimental EDS data. The comparison shows that the mean squared error between the simulated curve and the EDS data is 2% at 60,000 steps, which is a relatively low error. This result corresponds to 168 hours of exposure, with each iteration representing 10.08 seconds and a cell size of 0.125 µm.

Response 4: Thank you for your comments. We understand that the resolution of the image may not have been ideal. We hope that once we continue to delve deeper into this research, we will be able to obtain better resolutions so that both the results and the reader are not misled. In addition, following up on your comments, new graphics were added to complement the results given by the model in comparison with those obtained experimentally, which are shown on lines 283 to 297. Conclusions were added on lines 303 to 306 and 312 to 316.

Round 2

Reviewer 3 Report

Comments and Suggestions for Authors

I appreciate the corrections made to the article titled "Cellular Automaton Simulation of Corrosion in 347H Steel Exposed to Molten Solar Salt at Pilot Plant Scale." I agree with the modifications, which significantly enhance the quality of the manuscript.

However, I have noticed that the references are still not formatted according to the journal's requirements. I kindly request a detailed review of this aspect to ensure compliance with the established editorial guidelines.

I remain available for any additional adjustments that may be needed.